# Dietary Interventions of Salmon and Silver Carp Phospholipids on Mice with Metabolic Syndrome Based on Lipidomics

**DOI:** 10.3390/cells11203199

**Published:** 2022-10-12

**Authors:** Hongbiao Chen, Yun Li, Ping Yi, Hui Cao, Qi Wang, Xiuju Zhao

**Affiliations:** 1Team of Neonatal & Infant Development, Health and Nutrition, NDHN, School of Biology and Pharmaceutical Engineering, Wuhan Polytechnic University, Wuhan 430023, China; 2Kindstar Global Precision Medicine Institute, Wuhan 430223, China; 3School of Food Engineering, Wuhan Polytechnic University, Wuhan 430023, China

**Keywords:** metabolic syndrome, colonic contents, salmon phospholipids, silver carp phospholipids, extensive lipidomic

## Abstract

The number of metabolic syndromes (MetS) is increasing, and a fish phospholipid diet can reduce the risk of MetS. In this study, the changes in lipid metabolism of colon contents were analyzed by extensive lipidomics in mice with metabolic syndrome by fish phospholipid diet, and mice were randomly divided into experimental groups with different diet types by establishing a MetS model. After 14 weeks, the mice were sacrificed and the serum and colon contents were collected. Ultra-high liquid phase tandem mass spectrometry was used for broadly targeted lipidomic analysis, and the qualitative and quantitative detection of lipid metabolism changes in the colonic contents of mice. Under the intervention of fish phospholipids, MetS mice were significantly inhibited, serum total cholesterol (TC) and triglycerides (TG) decreased, serum high-density lipoprotein (HDL-C) and low-density lipoprotein (LDL-C) levels were improved, fasting blood glucose and insulin levels decreased, and inflammatory factors decreased. Through screening, it was found that thirty-three lipid metabolites may be key metabolites and five have significantly changed metabolic pathways. Modularizing lipid metabolites, it is possible to understand the extent to which different types and concentrations of fish phospholipids affect metabolic syndrome. Therefore, our study may provide new therapeutic clues for improving MetS.

## 1. Introduction

MetS is a pathological state of metabolic disorders, such as proteins, fats, carbohydrates, and other substances in the human body and is a complex metabolic disorder syndrome. As defined in the National Cardiopulmonary and Blood Institute and the American Heart Association Consensus Statement, metabolic syndromes are a combination of three or more risk factors, including abdominal obesity, hypertriglycerides, LDL and HDL cholesterol, hypertensive force, and elevated fasting blood glucose [1].

Nowadays, diet is considered an important tool for promoting health and preventing disease. Phospholipids play an important role in the body’s metabolic cycle, which can promote fat metabolism, improve immunity, and enhance the body’s ability to resist diseases. Theoretically, the function of phospholipids to promote fat metabolism can accelerate the metabolism of free fatty acids in peripheral blood and reduce the content of free fatty acids, thereby preventing and improving metabolic syndrome caused by central obesity or caused by the increase in peripheral blood free fatty acids. An increase in the activity of phospholipid transferable proteins has been found to be associated with obesity and human MetS [2]. Phospholipid transferable protein activity affects phospholipid transport, indicating the intervention effect of phospholipids on MetS. Due to the rich content of omega-3 fatty acids in fish oil, it is increasingly used to treat cardiovascular diseases [3]; fish phospholipids have received extensive clinical attention because of their pharmacological activity affecting various diseases and are increasingly used for the prevention and treatment of chronic inflammation, cardiovascular disease, and cancer in humans. A diet with pumpkin seed oil can modulate a number of pathways that reduce the fat load on the liver to alleviate MetS [4,5]. This means that there may be certain substances in the oil that can affect metabolic syndrome. Salmon and catfish phospholipids are rich in N-3 polyunsaturated fatty acids (PUFAs) [6,7,8], because n-3 polyunsaturated fatty acids (PUFAs) can inhibit triglyceride levels and inhibit metabolic disorders, such as cardiovascular disease and obesity, including dyslipidemia and insulin resistance, and it has been reported that the intake of n-3 polyunsaturated fatty acids (PUFAs) in the diet containing fish has a risk-reducing effect on the prevention of cardiovascular and cerebrovascular diseases, cancer, metabolic syndrome, and Parkinson’s syndrome [9,10]. Lipidomics is an emerging method for the comprehensive and systematic studies of multiple lipids [11].

Lipidomics has been applied to the reported biochemical mechanism of lipid metabolism in the animal models of hyperlipidemia [11,12]. There have been many studies analyzing the biological samples of blood and urine from mice with metabolic syndrome [13], but there is little knowledge about the intestinal content lipidomes. The mechanism of dietary changes in lipid profiles of intestinal contents in mice needs to be clarified. Therefore, in this study, fish phospholipids were used to intervene in high-fat mice, and ultra-performance liquid chromatography-tandem mass spectrometry (UPLC-MS/MS) was used to study the differently expressed lipid metabolites in intestinal contents between the high-fat group and the intervention group. Our research may provide clues as to the treatment of the metabolic diseases [14,15,16].

Due to the large number of lipid types, our extensive lipidomics analysis may reveal key metabolites or key metabolic pathways that affect metabolic syndrome, which may provide effective clues for the treatment of metabolic syndrome.

## 2. Materials and Methods

### 2.1. Animal Experiments and Sample Collection

The animal experiment was carried out according to the guidelines of the government of the People’s Republic of China and was approved by the Wuhan Polytechnic University Ethics Committee. Forty specific pathogens free (SPF) grade C57BL/6J male mice (about 4 weeks old) were purchased from the Experimental Animal Center of China Three Gorges University (license SCXK (Hubei) 2017-0012). All animals had free access to food and water and were housed in a room with relative humidity of 50 ± 10% and temperature at 23 ± 2 °C. After one week of adaptive breeding, all mice were randomly divided into low fat group (LF) groups (*n* = 10) and high fat (HF) groups (*n* = 30). Mice in the LF group were fed on a low-fat diet and mice in the HF group were fed on a high-fat diet. Mouse body weight and food intake were recorded weekly during the experiment. After establishing the MetS model, the mice in the HF group were randomly divided into the HF group (*n* = 10) and 4% silver carp (HM4) group (*n* = 5), 2% silver carp (HM2) group (*n* = 5), 2% salmon (SA2) group (*n* = 5), and 4% salmon (SA4) group (*n* = 5). The HF group was fed a high-fat diet, the HM4 group was fed a high-fat diet of 4% silver carp phospholipids, the HM2 group was fed a high-fat diet of 2% silver carp phospholipids, the SA4 group was fed a high-fat diet containing 4% salmon phospholipids, and the SA2 group was fed a high-fat diet of 2% silver carp phospholipids (Appendix A). After 14 weeks, serum was collected for clinical blood biochemical measurements, colon contents were collected and stored in liquid nitrogen freezing for lipidomic analysis.

### 2.2. Chemicals and Instruments

Methyl tert-butyl ether, acetonitrile, isopropanol, and methanol was purchased from Merck (Darmstadt, Germany). Dichloromethane and ammonium formate was purchased from Fisher. Formic acid was purchased from Sigma-Aldrich (Shanghai, China). Lipid Standard 12:0 Lyso PC, Cer (d18:1/4:0), and PC (13:0/13:0) was purchased from Avanti Polar Lipids (Alabaster, AL, USA). Ultrapure water is obtained through the Milli-Q system.

### 2.3. Biochemical and Histological Examination

#### 2.3.1. Biochemical Analysis

The biochemical factors of serum were analyzed using an automated biochemical analyzer (Olympus AU640, Tokyo, Japan).

#### 2.3.2. Histological Evaluation

Formalin-fixed liver tissue was embedded in paraffin, sliced into thin slices (3–4 μm), and stained. This assay was performed with hematoxylin and aeson (H&E), followed by microscopic evaluation.

### 2.4. UPLC-MS/MS Lipidomics Detection

#### 2.4.1. Sample Preparation

Three samples were randomly selected from each group for lipidomics detection. Colonic contents were thawed on ice. Of one sample, 20 mg was homogenized with 1 mL of mixture (include methanol, MTBE, and internal standard mixture) and a steel ball. The steel ball was removed, and the mixture was stirred for 15 min. An amount of 200 µL of water was added and stirred into the mixture for 1 min, which was then centrifuged with 13,000× *g* at 4 °C for 10 min. Of the supernatant, 300 µL was extracted and concentrated. The powder was dissolved with 200 µL mobile phase B, then stored in −80 °C. Finally, the dissolving solution was placed into a sample bottle for UPLC-MS/MS analysis.

#### 2.4.2. LC Conditions

The sample extracts were analyzed using widely targeted lipidomics with an UPLC-ESI-QTRAP-MS/MS system [15].

The analytical conditions were as follows—UPLC: column, Thermo Accucore™ C30 (2.6 μm, 2.1 mm × 100 mm); solvent system, A: acetonitrile/water (60/40, *v*/*v*, 0.1% formic acid, 10 mmol/L ammonium formate), B: acetonitrile/isopropanol (10/90, 0.1% formic acid, 10 mmol/L ammonium formate); the mobile phase gradient, A/B(80:20, *v*/*v*) at 0 min, 70:30 *v*/*v* at 2.0 min, 40:60 *v*/*v* at 4 min, 15:85 *v*/*v* at 9 min, 10:90 *v*/*v* at 14 min, 5:95 *v*/*v* at 15.5 min, 5:95 *v*/*v* at 17.3 min, 80:20 *v*/*v* at 17.3 min, 80:20 *v*/*v* at 20 min. The flow rate was 0.35 mL/min, and the column temperature was 45 °C; injection volume: 2 μL.

#### 2.4.3. Conditions of Tandem Mass Spectrometry

The effluent was connected to an ESI-triple quadrupole-linear ion trap (QTRAP)-MS/MS system. LIT and triple quadrupole (QQQ) scans were acquired on a triple quadrupole-linear ion trap mass spectrometer (QTRAP), QTRAP^®^ LC-MS/MS System, equipped with an ESI Turbo Ion-Spray interface, operating in positive and negative ion modes and controlled by Analyst 1.6.3 software (Sciex, Framingham, MA, USA). The ESI source operation parameters were as follows: ion source, turbo spray; source temperature 500 °C; ion spray voltage (IS) 5500 V (positive)–4500 V (negative); ion source gas 1 (GS1), gas 2 (GS2), curtain gas (CUR) was set at 45, 55, and 35 psi, respectively; the collision gas (CAD) was medium. Instrument tuning and mass calibration were performed with 10 and 100 μmol/L polypropylene glycol solutions in QQQ and LIT modes, respectively. QQQ scans were acquired as multiple reaction monitoring (MRM) experiments with collision gas (nitrogen) set to 5 psi. DP and CE for individual MRM transitions were optimized. A specific set of MRM transitions were monitored for each period according to the metabolites eluted within this period.

### 2.5. Qualitative and Quantitative Analysis of Lipidomics Data

Quality control (QC) was a representative “mean” sample containing all intestinal contents sample, it was included every 10 samples and used to calculate reproducibility. Based on the metabolic database, the metabolites of the samples were quantitatively and qualitatively analyzed by mass spectrometry. The signal intensity of characteristic ions was obtained in the detector, and the chromatographic peak was integrated and corrected with multiquant software 3.0.3 (Sciex, Framingham, MA, USA). The peak area of each chromatographic peak represented the relative content of the corresponding metabolite. This analysis of data processing was conducted in the software Analyst 1.6.3.

### 2.6. Statistical Analysis

Normalized signal intensities of metabolites were unit variance scaling. Unsupervised principal component analysis (PCA) and supervised orthogonal partial least squares (OPLS) were carried out by R package. The results of stratified cluster analysis (HCA) and Pearson correlation coefficient (PCC) of samples and metabolites are presented in the form of heat maps. Both HCA and PCC are performed by R-pack complex heat maps, and in order to visualize the metabolic pathways affected by metabolites, metabolomics pathway analysis is performed. Significantly related pathways were identified through enrichment analysis and pathway analysis (https://www.kegg.jp/, accessed on 11 March 2020) based on the Kyoto Gene and Genome Encyclopedia (KEGG).

## 3. Results

### 3.1. Results of Biochemical and Pathological Measurement

The weight gain of mice in the HF group was significantly higher than that in the normal control group, and after feeding, the weight of the mice decreased significantly, indicating that the accumulation of fat in HH mice was significantly inhibited, and the weight gain in the SA2, SA4, HM2, and HM4 groups decreased by 25%, 21%, 31%, and 21%, respectively, compared with the high-fat model group; The weight of adipose tissue decreased by 52.94%, 29.41%, 23.53%, and 29.41% respectively. The weight of the upper adipose tissue in the upper part of the sperm nest decreased by 13.73%, 3.92%, 0.98%, and 0.98%, respectively. HM was effective in reducing weight gain in mice in the high-fat model group and HM2 was better. SA functioned well in reducing perineal fat and upper cell fat in high-fat mice but SA2 worked best. Compared with the normal control group, the high-fat diet led to a decrease in the heart and brain index of the mice, which improved somewhat after the addition of fish phospholipids, and slightly better effects of low-dose SA and HM were seen (Table 1).

The serum AST and ALT content of mice in the HM group was significantly higher than that in the normal control group, indicating that the liver function of the mice in the HM group had been severely impaired and the serum AST and ALT levels of each phospholipid group were significantly reduced compared with the HM group, the SA effect was better than that of silver carp phospholipids, the addition effect of SA and HM low doses was more significant, and the serum AST and ALT levels of mice in the SA2 group decreased by 21.55% and 65.46%, respectively, showing the optimal effect. Compared with the normal control group, the serum TC, TG and LDL-C levels in the HF group were significantly increased, and the HDL-C level in the serum was reduced, and the serum TC levels in the SA2, SA4, HM2, and HM4 groups decreased by 10.04%, 10.03%, 19.70%, and 19.52%, respectively. TG levels decreased by 7.59%, 24.05%, 4.43%, and 18.35%, respectively, and the overall performance of phospholipids with 4% content was better than that of 2% phospholipids and the 4% salmon phospholipid group had the best effect. High doses of phospholipids showed slightly better results in reducing LDL-C with 4% SA phospholipids being the most effective (Table 2).

HF mice’s fasting blood glucose level, histiocyte insulin level, and adiponectin levels were significantly higher than the normal control group; blood glucose was significantly abnormal, but after the addition of phospholipids, blood glucose levels were significantly lower than the HF group and a high concentration of phospholipid addition to the mouse fasting blood glucose level was even better—histiocyte insulin level decreased, indicating that it can improve the insulin function damage caused by high-fat diet to a certain extent, while 4% SA significantly reduced the insulin level in high-fat mice. Adiponectin levels in the phospholipid group were also significantly improved in the higher lipid model group, the 4% SA group had the most increased adiponectin levels, and 4% SA showed the best effect in improving insulin resistance in high-fat mice. Compared with the normal control group, the cytoinflammatory factors MCP-1, IL-6, and TNF-α of HF were significantly increased by 33.26%, 47.59%, and 41.76%, respectively, indicating that the mice’s long-term high-fat diet led to the occurrence of inflammatory reactions in vivo, but after the addition of SA and HM, the level of inflammatory factors was significantly reduced, and HM showed better results than SA. HM was better than SA at reducing IL-6 in high-fat mice. HM2 lowered the level of TNF-α in high-fat mice more significantly (Table 3).

The liver cells of the mice in the normal control group were normal in morphology, the liver cells were arranged radially with the central vein as the center, and there were no fat vacuoles; however, a large number of fat vacuoles of different sizes appeared in the liver cells of the mouse in the high-fat model group, the cell body was swollen, and the nucleus was offset from the cell center, indicating that the long-term high-fat diet led to liver lesions in mice. However, after the addition of salmon and silver carp phospholipids, the liver steatosis decreased, the fat vacuoles became less, the cell volume was normal relative to the high-fat model group, the effect of phospholipid additions was not much different, and the effect of salmon phospholipids was slightly better (Appendix A).

### 3.2. QC for Principal Component Analysis

We found that the Quality control (Mix samples) were closest to the origin, which meant that our sample had high quality and experimental results that were relatively reliable, and the variance of each group on the PC1 was very small but the variance of each group on the PC2 was big (Figure 1).

### 3.3. S-Plot

The S-plots between HF and the other group (LF, HM2, HM4, SA2, and SA4 each) displayed significant differences in lipid metabolites (Figure 2). We further analyzed the modules of these different lipids using correlation and cluster heatmaps.

### 3.4. Correlation Analysis

The relevant heatmaps between HF and several other groups (LF, HM2, HM4, SA2, and SA4) (Figure 3) can be divided into six modules. The first module was mainly of the fatty acyl (FFA) subclass, and there were 18, 17, 20, 12, and 19 different metabolites in different comparison groups, respectively. The second module was mainly glycerophospholipid (GP) and sphingolipid (SL) classes (mainly phosphatidylcholine (PC), phosphatidylethanolamine (PE), phosphatidylglycerol (PG) and ceramide subclasses); the number of differential metabolites in different comparison groups was 49, 23, 15, 26, and 22, respectively. The third module consisted mainly of the glyceride (GL) class and DG subclass; the number of differential metabolites in different comparison groups was 31, 14, 26, 18, and 28, respectively. The fourth module was mainly GP class (PC and PE subclasses); the number of differential metabolites in different comparison groups was 74, 20, 22, 40, and 42, respectively. The fifth module was mainly SL class, sphingomyelin (SM) subclass; the number of differential metabolites in different comparison groups was 7, 20, 12, 7, and 16 respectively. The sixth module was mainly the GL class and TG subclass; the number of differential metabolites in different comparison groups was 80, 42, 38, 62, and 36, respectively.

### 3.5. Data Analyse

This study found that the standards for lipid metabolites with differential expression were FC ≥ 2 or ≤0.5 and VIP ≥ 1. As shown in Figure 4, 259 significantly expressed lipids (247 down-regulating and 12 up-regulating) were screened in the LF and HF groups, 133 significantly expressed lipids (34 down-regulating and 99 up-regulating) were screened in the HM4 and HF groups, 136 significantly expressed lipids were screened in the HM2 and HF groups (58 down-regulating and 78 up-regulating), and 163 significantly expressed lipids (28 down-regulating and 135 up-regulating) were screened in the SA4 and HF groups. Of these, 165 significantly expressed lipids (25 down-regulating and 140 up-regulating) were screened in the SA2 and HF groups (Figure 4).

### 3.6. Analysis of Venn Results

After selecting, we obtained 461 different lipids (Appendix A), this situation is represented by a Venn diagram. Some lipids are expressed in only one case, and some lipids are expressed in all cases (Figure 5).

For the HM group, we found that 63 lipids were expressed simultaneously in the LF, HF, HM2, and HM4 groups (five major classes, ten subclasses) and for SA, we found that 62 lipids were expressed simultaneously in the LF, HF, SA2, and SA4 groups (five major classes, seven subclasses) (Table 4 and Appendix A).

We found 33 lipids with significant effects were included under all experimental conditions (Table 5).

### 3.7. Analysis of Kmeans Clustering

In order to study the variation trend of the relative contents of metabolites in different sample species, we pre-divided the data into 461 lipids. These data were divided by the relative content of differential metabolites and we obtained 12 means clusters after standardization and centralization. We found that thirty-three lipids that had significant effect in all situations in almost all five clusters (Figure 6), eleven lipids (one DG and ten TG) were found in cluster 1, two lipids (two SM) were found in cluster 3, ten lipids (three DGs and seven TGs) were found in cluster 6, four lipids (three DGs and one TG) were found in cluster 9, and six lipids (three FFAs and three DGs) were found in group 10. We found that the normalized intensity in cluster 1 grew slowly in HF to SA4 and then slowly decreased to HM4 in SA2, but the normalized intensity of HM4 was higher than that of HF, and LF had the highest normalized intensity growth trend and growth. We found that the normalized strength of cluster 6 grew slowly in HF to SA2, significantly in SA2 to SA4, then significantly lowered in SA4 to HM2, and slowly decreased in HM2 to HM4, but then significantly increased in HM4 to LF, with LF having the highest normalized intensity. We found that the normalized intensity of cluster 10 increased significantly in HF to SA4, sa4 had the highest normalized intensity, and then decreased significantly in SA4 to HM2, but grew slowly in HM2 to LF.

### 3.8. Analysis of KEGG

We identified 33 lipids that had significant effects under all experimental groups and used their KEGG IDs to discover their relationships (https://www.metaboanalyst.ca/, accessed on 11 August 2022). The corresponding bubble chart and histogram are obtained. There are mainly three different lipid classes, namely fatty acids, conjugates, phospholipids, and glycerides, each with a different enrichment ratio and *p*-value. Fatty acids and conjugates have the highest enrichment rates, with *p*-values close to zero; Glycerol has the smallest enrichment ratio, and its *p*-value is close to zero 6 (Figure 7A,B). There are also five different pathways associated with phospholipid interventions, namely glycerol metabolism, sphingolipid metabolism, phosphatidyl inositol signaling system, inositol phosphate metabolism, and glycerol phospholipid metabolism (Figure 7C,D).

## 4. Discussion

MetS was closely related to free fatty acids [17]. According to Juliana Bermudez-Cardona’s research. MetS patients had significantly greater total free fatty acids than average person [18]. MetS was also closely related to central obesity, as after the formation of visceral fat, adipose cells in the body begin rapidly lipolyze, and excessive free fatty acids were oxidized and decomposed in muscle to affects the muscle’s use of glucose [19].

Lipids are known to be one of major sources of food energy and play a crucial role in the systemic metabolism [20]. Phospholipids are one of the most diverse human lipid classes [21]. According to our research, the main active ingredients of silver carp phospholipids and salmon phospholipids may be DG and SM, respectively.

In the correlation analysis and cluster analysis, we found that differences in the intestinal contents of lipid metabolites mainly belonged to three classes, namely SL, GP, and GL. In the cluster analysis, the SL and GP classes (sphingomyelin, PG, PE, and PC subclasses) were up-regulated into the HF group, while the GL class (DG and TG subclasses) were up-regulated in the LF group and the intervention groups.

Significant increases in sphingolipids may reduce reverse cholesterol transport and thereby increase the risk of hyperlipidemia-related diseases [22,23]. Sphingomyelin is the main component of the cell membrane. It has been reported that sphingomyelinases hydrolyze sphingomyelin, leading to the release of ceramide and the accumulation of phytosphingosine [24]. The phosphatidylinositol signaling system pathway suggested that sphingomyelins were involved in the regulation of cell proliferation and apoptosis through ceramides. PGs are inflammation responsive lipids indirectly regulated by the gut microbiota via endotoxins and regulate adipose tissue homeostasis in obesity.

In addition, PGs are related to the severity of obesity in mice and humans [25]. PE and its component, ethanolamine, have cholesterol-lowering effects that can be attributed to the increased fecal excretion of neutral steroids [26]. PC is by far the most abundant dietary source of choline in most humans [27]. Studies have shown that plasma PC in MetS patients, which is also an important biomarker of T2DM, is significantly reduced and suggests that the metabolic pathway of PC is closely associated with MetS [28,29]. PC has the effects of delaying cell senescence, inducing adipocyte cell death to reduce local fat deposition, reducing chronic inflammation in adipose tissue, and promoting the conduction of insulin signaling pathways to prevent insulin resistance [30,31,32]. The intestinal contents of sphingomyelin, PG, PE, and PC subclasses were significantly up-regulated in the HF group, indicating that their levels in plasma would be reduced, which would aggravate the metabolic syndrome. The plasma cholesterol level was significantly increased in our diet-induced hyperlipidemic mice. This is due to the increased intestinal absorption of cholesterol, diminished phytosterol absorption, and reduced hepatic campesterol [12].

DG had the function of reducing the TG content in blood and reducing fat accumulation in the liver of mice that were fed a high fat diet [15]. DG can reduce the level of blood lipids and alleviate the disturbance of lipid metabolism in animals by reducing the production of lipid peroxidation products and reduce glucose toxicity in mice kidneys [20]. In this paper, the DG and TG of intestinal contents were up-regulated in both the control and intervention groups, indicating that the intestinal absorption of TG and DG decreased, so the serum TG and DG decreased correspondingly. There are numerous studies that have shown it. N-3 PUFA phospholipids improve metabolic syndrome by altering liver gene expression, accelerating fatty acid metabolism, reducing inflammatory response, and enhancing insulin sensitivity [33,34]. So, the improvement of fish phospholipids for metabolic syndrome may be achieved through n-3 PUFA. This suggests that eating fish phospholipids can improve MetS in the lipid metabolism of the intestinal contents of mice; however, the specific mechanism needs to be further elucidated. In the correlation heatmap analysis, the differential metabolites of modules 2, 3, 4, and 6 were mainly SL, GP, and GL classes, and the number of differential metabolites was significantly reduced in the intervention groups. These results indicated that the fish phospholipid intervention could effectively reduce the number of differential metabolites of GP, SL, and GL classes, thus effectively improving the metabolic syndrome. The six correlation modules should be verified as to whether these modules are efficient in other metabolic diseases and other nutrition interventions using widely targeted lipidomics.

## 5. Conclusions

In this paper, the dietary intervention effects of different types and concentrations of fish phospholipids in mice with metabolic syndrome were studied, and the lipid content in each group of colonic contents was determined by ultra-high liquid phase and mass spectrometry, and lipidomics analysis was performed regarding the intervention of fish phospholipids. The results found that the metabolic syndrome improved, and there are two mechanisms of possible improvement: One is that the n-3 PUFA contained in silver carp and salmon phospholipids can greatly reduce the abnormal accumulation of fat in high-fat mice and increase the level of adiponectin secretion in high-fat mice, improving insulin resistance and thus improving metabolic syndrome. Second, silver carp and salmon phospholipids contain n-3 PUFA, which can significantly reduce the serum IL-6, TNF-α, and MCP-1 levels of high-fat mice, inhibiting the body’s inflammatory response and thus improving insulin resistance and metabolic syndrome.

The study found that 33 metabolites were significantly expressed in each group and may become key metabolites affecting metabolic syndrome, with significant changes in five metabolic pathways. The intestinal contents are divided into six related modules and two cluster modules. Due to the different symptoms of metabolic syndrome, the functions of different fish phospholipids is not the same, and by uncovering the effects of different fish phospholipids on metabolic syndrome and the common lipids and the differences in the treatment of various fish phospholipids and by looking for key metabolites, metabolic pathways, similar nutrients, and different types and concentrations of phospholipids on metabolic equivalent intervention, symptomatic treatment becomes conducive, thus helping to find the best treatment plan and new treatment methods, which also provide clues for the search for new biomarkers. Therefore, the results of this study may provide new therapeutic ideas for effectively reducing the risk of metabolic syndrome.

## Figures and Tables

**Figure 1 cells-11-03199-f001:**
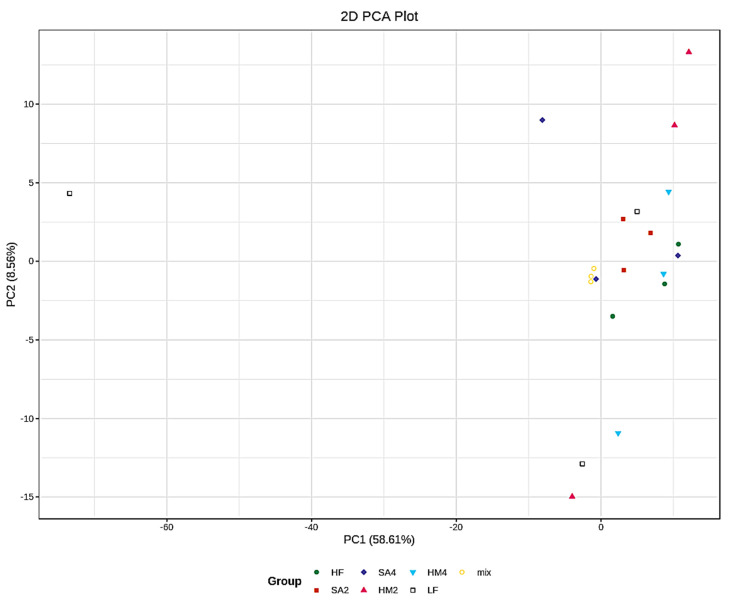
The abscissa represents the first principal component, the ordinate represents the second principal component, and the percentage represents the value of the contribution of the principal component to the sample difference. Each point in the figure represents a sample, and samples from the same group are represented. Each point in the figure represents a sample, and samples from the same group are represented using the same color.

**Figure 2 cells-11-03199-f002:**
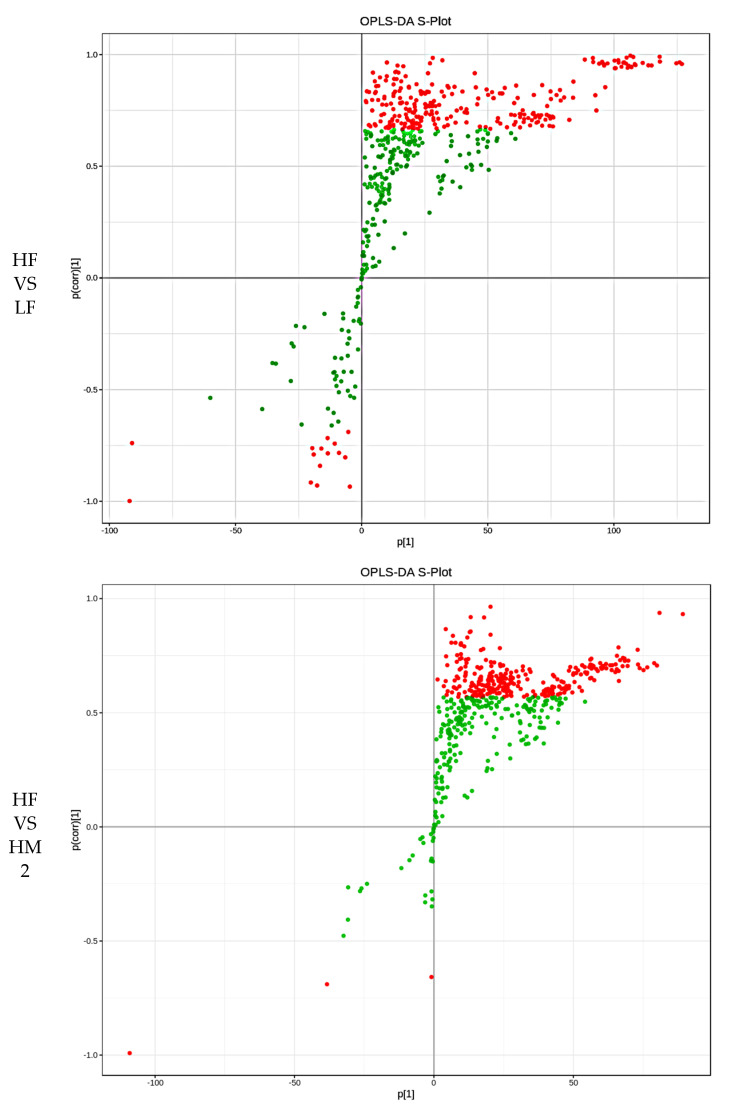
The abscissa indicates the co-correlation coefficient between the principal components and metabolites, the ordinate indicates the correlation coefficient between the principal components and metabolites, the closer to the upper right corner and the lower left corner of the metabolite indicates that the difference is more significant, the red dot indicates that the VIP value of these metabolites is greater than or equal to 1, and the green dot indicates that the VIP value of these metabolites is less than 1.

**Figure 3 cells-11-03199-f003:**
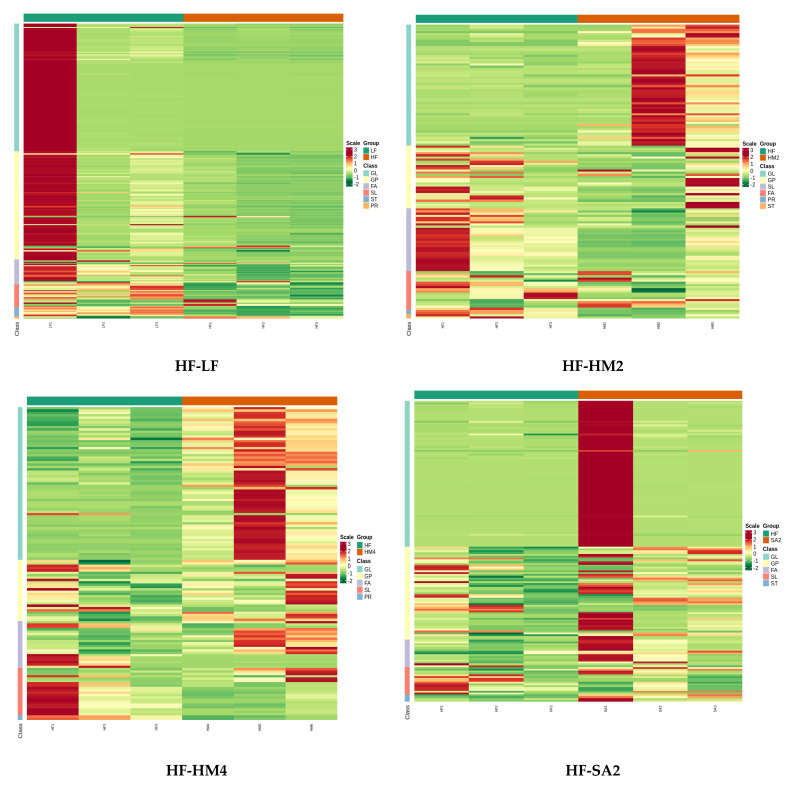
The horizontal is the sample information, the longitudinal is the metabolite information, the cluster tree on the left side of the figure is the metabolite cluster tree, the scale is the expression obtained after the standardized treatment (the higher the expression of the color, the higher the redness), and the group is the grouping.

**Figure 4 cells-11-03199-f004:**
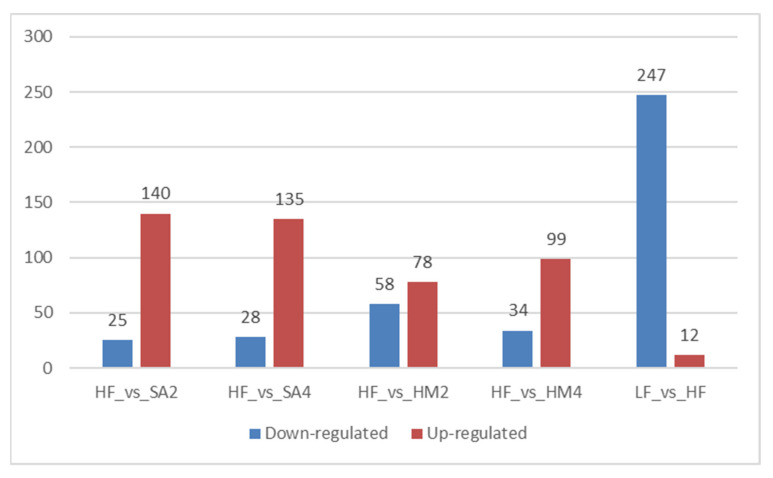
The abscissa represents the comparison of the HF group with each group of metabolites, and the ordinate represents the count of significantly changed metabolites; the red part indicates up-regulating and the blue part indicates down-regulating.

**Figure 5 cells-11-03199-f005:**
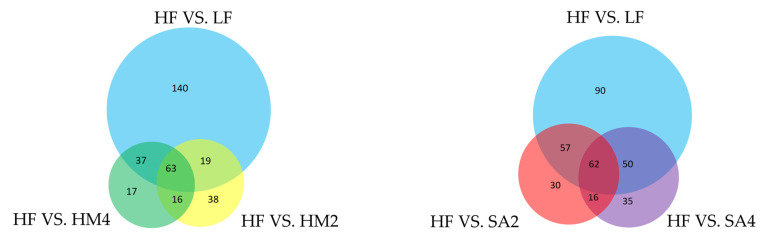
Each circle in the figure represents a comparison group, the numbers of circles and overlapping parts of the circle represent the number of differential metabolites common between the comparison groups, and the numbers without overlapping parts represent the number of differential metabolites specific to the comparison group.

**Figure 6 cells-11-03199-f006:**
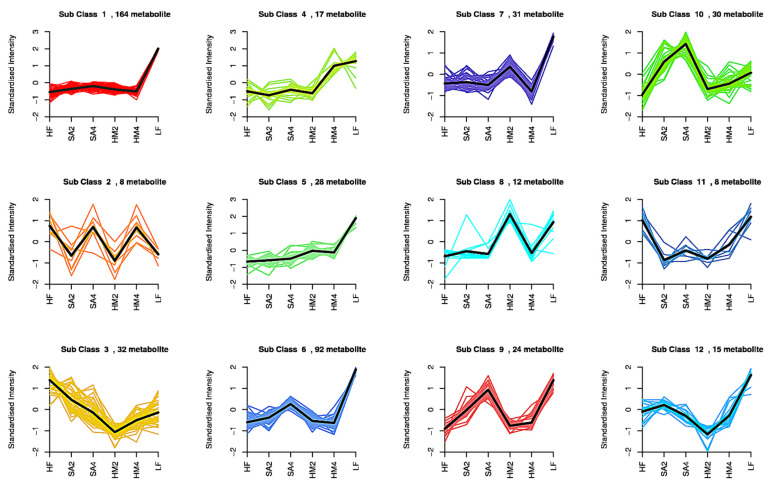
Overview of the K-means. The 461 lipids have been divided 12 means clusters. The abscissa represents different conditions, and the ordinate indicates different standardized intensity. The black line means the trend line in the group. Subclass in figures indicate means cluster.

**Figure 7 cells-11-03199-f007:**
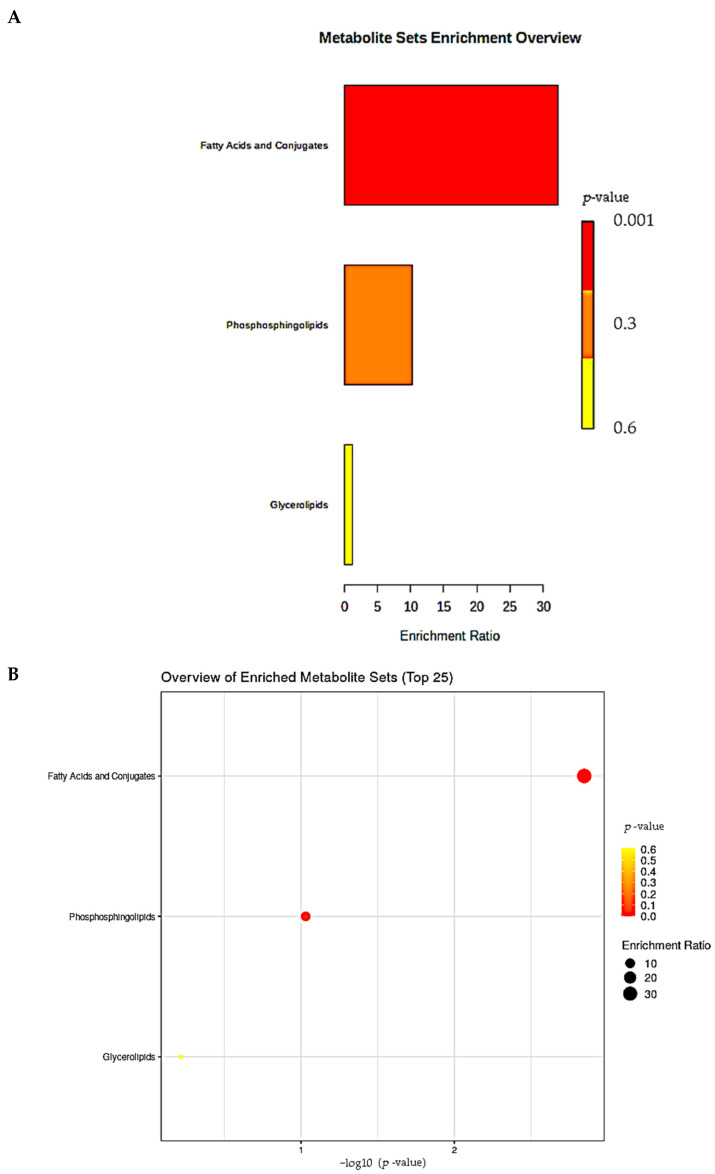
Overview of the main classes and pathways of KEGG. (**A**) KEGG subclass categories of differential metabolites. (**B**) KEGG subclass enrichment of differential metabolites. (**C**) KEGG pathway categories of differential metabolites. (**D**) KEGG pathway enrichment of differential metabolites. The abscissa represents the rich factor corresponding to each main class. The ordinate is the class name. The colorbar is *p* value. The redder it is, the more significant.

**Table 1 cells-11-03199-t001:** Effects of phospholipids on growth indexes of hyperlipidemia mice.

Index	HF	2% SA	4% SA	2% HM	4% HM	LF
Food intake (g/mouse/d)	3.3 ± 0.02 ^b^	3.1 ± 0.12 ^c^	2.7 ± 0.11 ^d^	3.2 ± 0.06 ^bc^	2.6 ± 0.09 ^d^	4.0 ± 0.04 ^a^
energy intake (kcal/mouse/d)	16.1 ± 0.07 ^a^	15.1 ± 0.59 ^b^	13.2 ± 0.54 ^c^	15.6 ± 0.27 ^ab^	12.5 ± 0.42 ^c^	15.5 ± 0.14 ^ab^
Original weight/g	16.8 ± 0.02 ^a^	16.5 ± 0.01 ^bc^	16.5 ± 0.10 ^bc^	16.4 ± 0.11 ^c^	16.6 ± 0.08 ^b^	16.4 ± 0.08 ^c^
Weight gain/g	10.0 ± 0.50 ^a^	7.5 ± 0.15 ^bc^	7.9 ± 0.31 ^b^	6.9 ± 0.36 ^c^	7.9 ± 0.42 ^b^	5.3 ± 0.76 ^d^
Liver/%	3.76 ± 0.03 ^b^	3.87 ± 0.04 ^b^	3.85 ± 0.03 ^b^	4.03 ± 0.03 ^a^	4.00 ± 0.10 ^a^	4.06 ± 0.13 ^a^
Kidney/%	1.12 ± 0.01 ^a^	1.21 ± 0.01 ^a^	1.13 ± 0.05 ^a^	1.18 ± 0.03 ^a^	1.12 ± 0.09 ^a^	1.20 ± 0.10 ^a^
Heart/%	0.54 ± 0.07 ^a^	0.63 ± 0.08 ^a^	0.62 ± 0.11 ^a^	0.60 ± 0.10 ^a^	0.65 ± 0.12 ^a^	1.20 ± 0.10 ^a^
Brain/%	1.70 ± 0.11 ^b^	1.79 ± 0.03 ^ab^	1.76 ± 0.08 ^ab^	1.89 ± 0.11 ^a^	1.71 ± 0.05 ^b^	1.90 ± 0.03 ^a^
perirenal fat/%	0.17 ± 0.03 ^a^	0.08 ± 0.02 ^b^	0.12 ± 0.03 ^b^	0.13 ± 0.01 ^ab^	0.12 ± 0.03 ^b^	0.09 ± 0.03 ^b^
Adipose tissue above testis/%	1.02 ± 0.13 ^a^	0.88 ± 0.09 ^ab^	0.98 ± 0.04 ^a^	1.01 ± 0.06 ^a^	1.01 ± 0.01 ^a^	0.74 ± 0.09 ^b^

Note: Different letters superscript on the same column indicate significant differences (*p* < 0.05).

**Table 2 cells-11-03199-t002:** Effects of phospholipids on serum indicators of high fat mice.

Group	AST/U/L	ALT/U/L	TC/mmol/L	TG/mmol/L	HDL-C/mmol/L	LDL-C/mmol/L
HF	108.7 ± 1.04 ^a^	49.8 ± 2.76 ^a^	5.38 ± 0.42 ^a^	1.58 ± 0.03 ^a^	2.67 ± 0.05 ^e^	1.62 ± 0.06 ^a^
2% SA	85.2 ± 2.44 ^c^	17.2 ± 0.76 ^d^	4.84 ± 0.47 ^ab^	1.46 ± 0.01 ^b^	2.78 ± 0.02 ^d^	0.98 ± 0.06 ^c^
4% SA	97.8 ± 1.56 ^b^	19.4 ± 0.61 ^cd^	4.87 ± 0.15 ^ab^	1.20 ± 0.05 ^d^	3.01 ± 0.02 ^b^	0.66 ± 0.08 ^e^
2% HM	98.2 ± 1.71 ^b^	21.5 ± 0.84 ^c^	4.32 ± 0.12 ^bc^	1.51 ± 0.01 ^b^	2.89 ± 0.01 ^c^	1.14 ± 0.03 ^b^
4% HM	101.1 ± 1.90 ^b^	30.0 ± 0.10 ^b^	4.33 ± 0.42 ^bc^	1.29 ± 0.02 ^c^	2.91 ± 0.01 ^c^	0.83 ± 0.01 ^d^
LF	82.5 ± 1.95 ^c^	17.4 ± 1.69 ^d^	4.01 ± 0.35 ^c^	1.10 ± 0.03 ^e^	3.16 ± 0.02 ^a^	0.41 ± 0.02 ^f^

Note: different superscript letters in the same column indicate significant difference (*p* < 0.05).

**Table 3 cells-11-03199-t003:** The effects of the phospholipid on high fat diet mice.

Group	FBG	Insulin	Adiponectin	MCP-1	IL-6	TNF-α
HF	8.4 ± 0.15 ^a^	304.12 ± 0.99 ^a^	753.12 ± 0.97 ^f^	222.69 ± 1.34 ^a^	501.02 ± 0.68 ^a^	689.13 ± 1.13 ^a^
2% SA	6.4 ± 0.10 ^bc^	301.26 ± 0.35 ^b^	951.34 ± 0.69 ^d^	207.24 ± 1.33 ^b^	482.72 ± 0.24 ^b^	632.26 ± 1.07 ^b^
4% SA	5.9 ± 0.21 ^c^	243.14 ± 1.22 ^c^	1251.38 ± 1.08 ^b^	197.16 ± 1.57 ^c^	382.51 ± 0.98 ^c^	599.87 ± 0.26 ^c^
2% HM	6.7 ± 0.15 ^b^	303.89 ± 1.10 ^a^	1165.91 ± 1.12 ^c^	171.21 ± 0.33 ^d^	265.32 ± 0.38 ^e^	429.87 ± 0.85 ^e^
4% HM	6.1 ± 0.56 ^c^	303.61 ± 0.73 ^a^	926.71 ± 1.44 ^e^	167.50 ± 1.87 ^e^	277.12 ± 1.04 ^d^	465.13 ± 0.08 ^d^
LF	5.1 ± 0.20 ^d^	102.33 ± 0.22 ^d^	1497.78 ± 0.79 ^a^	148.62 ± 1.12 ^f^	262.58 ± 1.40 ^f^	401.34 ± 1.19 ^f^

Note: FBG (fasting blood-glucose) the measurement unit of fasting blood glucose is mmol/L and other is pg/mL. Different letters superscript on the same column indicates significant differences (*p* < 0.05).

**Table 4 cells-11-03199-t004:** Lipid species of metabolites in colonic contents of mice in the LF group, the HF group, the HM2 group, the HM4 group, the SA2 group, and the SA4 group.

Class	Subclass	Metabolites Detected	LF vs. HF	HF vs. SA4, HF vs. SA2, LF vs. HF	HF vs. HM4, HF vs. HM2, LF vs. HF
Glycerophospholipids	PC/PC-O	80	40	10	1
	PG	5	2	0	2
	PS	11	10	0	2
	PE/PE-P	71	30	3	
	LPC/LPC-O	27	4	0	
	LPG	1			
	LPE	18	1	0	
	LPS	3	2		1
	PA	7	6	0	
	PI	1			
	LPA	2			
	LPI	1			
Glycerolipids	DG	48	31	16	14
	MG	3	1	0	
	TG	93	80	22	32
Sphingolipids	SM	24	7	2	4
	Cer/ Cer-p	33	13	0	
Isopentenol lipids	COQ	2	2		2
Sterol esters	CE	14	8	1	
Fatty Acyls	CAR	38	4	0	
	Eicosanoid	18	4	0	1
	FFA	38	14	8	4
Total	22	538	259	62	63

Note: PC/PC-O (phosphatidycholine), PG (phosphatidylglycerol), PS (phosphatidylserine), PE/PE-P (phosphatidylethanolamine), LPC/LPC-O (lysophosphatidycholine), LPG (lysophosphatidylglycerol), LPE (lysophosphatidylethanolamine), LPS (lysophosphatidylserine), PA (phospholipid acid), PI (lysophosphatidylinositol), LPA (lyaophospholipid acid), LPI (phosphatidylinositol), DG (Diglycerides), MG (Monoglycerides), TG (triglycerides), SM (sphingomyelin), Cer/Cer-p (ceramide), COQ (Coenzyme Q), CE (cholesterol ester), CAR (acyl carnitine), FFA (free fatty acid).

**Table 5 cells-11-03199-t005:** Overview of the 33 lipids.

Compounds	Main Class	Sub Class	Cpd_ID	Cluster Class	Kegg_Map
FFA (19:1)	FA	FFA	-	10	--
FFA (18:2)	FA	FFA	C20388	10	--
FFA (18:3)	FA	FFA	C08364	10	--
DG (18:0/18:2/0:0)	GL	DG	C00641	9	ko00561, ko00562, ko00564, ko01100, ko04070, ko04730
DG (18:2/20:0/0:0)	GL	DG	C00641	6	ko00561, ko00562, ko00564, ko01100, ko04070, ko04730
DG (16:0/18:3/0:0)	GL	DG	C00641	6	ko00561, ko00562, ko00564, ko01100, ko04070, ko04730
DG (18:2/20:1/0:0)	GL	DG	C00641	9	ko00561, ko00562, ko00564, ko01100, ko04070, ko04730
DG (18:1/18:3/0:0)	GL	DG	C00641	10	ko00561, ko00562, ko00564, ko01100, ko04070, ko04730
DG (18:2/18:3/0:0)	GL	DG	C00641	9	ko00561, ko00562, ko00564, ko01100, ko04070, ko04730
DG (16:1/20:4/0:0)	GL	DG	C00641	6	ko00561, ko00562, ko00564, ko01100, ko04070, ko04730
DG (18:2/20:4/0:0)	GL	DG	C00641	10	ko00561, ko00562, ko00564, ko01100, ko04070, ko04730
DG (16:0/22:6/0:0)	GL	DG	C00641	1	ko00561, ko00562, ko00564, ko01100, ko04070, ko04730
DG (18:2/22:4/0:0)	GL	DG	C00641	10	ko00561, ko00562, ko00564, ko01100, ko04070, ko04730
SM (d20:0/24:1)	SL	SM	C00550	3	ko00600, ko01100, ko04071, ko04217
SM (d18:1/26:0)	SL	SM	C00550	3	ko00600, ko01100, ko04071, ko04217
TG (16:0/16:0/18:1)	GL	TG	C00422	1	ko00561, ko01100, ko04714, ko04923, ko04931, ko04975, ko04977, ko04979
TG (14:0/18:0/20:1)	GL	TG	C00422	6	ko00561, ko01100, ko04714, ko04923, ko04931, ko04975, ko04977, ko04979
TG (18:0/18:0/18:1)	GL	TG	C00422	9	ko00561, ko01100, ko04714, ko04923, ko04931, ko04975, ko04977, ko04979
TG (16:0/16:1/18:1)	GL	TG	C00422	1	ko00561, ko01100, ko04714, ko04923, ko04931, ko04975, ko04977, ko04979
TG (16:0/16:1/20:1)	GL	TG	C00422	6	ko00561, ko01100, ko04714, ko04923, ko04931, ko04975, ko04977, ko04979
TG (14:0/18:0/20:2)	GL	TG	C00422	6	ko00561, ko01100, ko04714, ko04923, ko04931, ko04975, ko04977, ko04979
TG (14:0/20:1/20:1)	GL	TG	C00422	1	ko00561, ko01100, ko04714, ko04923, ko04931, ko04975, ko04977, ko04979
TG (14:0/18:1/20:2)	GL	TG	C00422	1	ko00561, ko01100, ko04714, ko04923, ko04931, ko04975, ko04977, ko04979
TG (14:0/18:2/20:1)	GL	TG	C00422	6	ko00561, ko01100, ko04714, ko04923, ko04931, ko04975, ko04977, ko04979
TG (16:0/16:1/20:2)	GL	TG	C00422	6	ko00561, ko01100, ko04714, ko04923, ko04931, ko04975, ko04977, ko04979
TG (18:0/18:1/18:2)	GL	TG	C00422	6	ko00561, ko01100, ko04714, ko04923, ko04931, ko04975, ko04977, ko04979
TG (14:0/18:1/20:3)	GL	TG	C00422	1	ko00561, ko01100, ko04714, ko04923, ko04931, ko04975, ko04977, ko04979
TG (14:0/18:2/20:2)	GL	TG	C00422	1	ko00561, ko01100, ko04714, ko04923, ko04931, ko04975, ko04977, ko04979
TG (14:0/16:0/22:4)	GL	TG	C00422	1	ko00561, ko01100, ko04714, ko04923, ko04931, ko04975, ko04977, ko04979
TG (18:1/18:1/18:2)	GL	TG	C00422	6	ko00561, ko01100, ko04714, ko04923, ko04931, ko04975, ko04977, ko04979
TG (18:1/18:2/18:2)	GL	TG	C00422	1	ko00561, ko01100, ko04714, ko04923, ko04931, ko04975, ko04977, ko04979
TG (18:0/18:2/18:3)	GL	TG	C00422	1	ko00561, ko01100, ko04714, ko04923, ko04931, ko04975, ko04977, ko04979
TG (18:2/18:2/18:2)	GL	TG	C00422	1	ko00561, ko01100, ko04714, ko04923, ko04931, ko04975, ko04977, ko04979

## Data Availability

Data generated or analyzed during this study are provided in full within the published article and its Appendix A.

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
