# Peer review of "Dietary Interventions of Salmon and Silver Carp Phospholipids on Mice with Metabolic Syndrome Based on Lipidomics"

_cells, 2022, doi:10.3390/cells11203199_

Round 1

Reviewer 1 Report

I have some specific comments on this MS. Please try to address these comments:

L16: use sacrifice instead of the word "Kill". 

Abstract:  include some important data in your abstract then it looks nice

L36: ....of liver to alleviate MetS.1,2 This means.... the way you cited all references are very peculiar. It should ....of liver to alleviate MetS [1,2]. This means...   Please make corrections for the rest of the MS. 

Introduction: research issues are not well-structured. Please rewrite this part and use more recent references to formulate a strong hypothesis for this study.

Introduction: a solid justification is required: why did you use fish phospholipids for this study?

L103: Histological evaluation required detail description

L184: table 3: for insulin, there should not be any significant differences between 304.12, 301.26, 303.89. 

Statical analysis: I would recommend analysing all the data presented in this MS and revising the results

Figures are not clear, nothing is visible to the naked eye. Readers will not use a microscope to see your figures 2&3

Author Response

Reviewer 1:

Comments and Suggestions for Authors

I have some specific comments on this MS. Please try to address these comments: L16: use sacrifice instead of the word "Kill".

I've changed "Kill" to sacrifice

Abstract:    include some important data in your abstract then it looks nice

L36: ....of liver to alleviate MetS.1,2 This means.... the way you cited all references are very peculiar. It should ....of liver to alleviate MetS [1,2]. This means...    Please make corrections for the rest of the MS.

I have re-formatted all the references cited in the article

Introduction: research issues are not well-structured. Please rewrite this part and use more recent references to formulate a strong hypothesis for this study.

Introduction: a solid justification is required: why did you use fish phospholipids for this study?

The introduction has been rewritten and references are given to prove why fish phospholipids have been explained and explained in the article, due to the rich content of omega-3 fatty acids in fish oil, it is increasingly used to treat cardiovascular disease, silver carp and salmon as a typical fish in freshwater fish and marine fish, rich in unsaturated fatty acids and phospholipids, and rich in a large amount of EPA and DHA, has a certain active research value

Unsaturated fatty acids can be a good improvement of cardiovascular and cerebrovascular diseases, especially DHA and EPA, can more effectively prevent cardiovascular diseases, reduce the probability of tumor diseases, in addition to anti-inflammatory, anti-allergic effects, and

regular consumption of products containing DHA and EPA can also improve heart function. So, in the end, fish phospholipids were chosen

L103: Histological evaluation required detail description Histological assessment has been re-described

L184: table 3: for insulin, there should not be any significant differences between 304.12, 301.26, 303.89.

Reanalysis calculations of the proposed section revealed discrepancies between 3 01.26 and

304.12 and 3 03.89 and therefore no modifications were made.

Statical analysis: I would recommend analysing all the data presented in this MS and revising the results

Figures are not clear, nothing is visible to the naked eye. Readers will not use a microscope to see your figures 2&3

Reviewer 2 Report

 In present study, researchers used an “extensive lipidomics analysis” to provide a better understanding of metabolic disorders and probable ways of curing or improving healing procedures by adding fish oil in routine diet.

This is one of the most complicated articles that I have ever read: there are many experimental groups which are compared to each other and to a large spectrum of lipids that have already been compared amongst each experimental group.

My first proposition to authors is to start simplifying the paper by the title.

Despite the complicated nature of “omics”, and they way of presenting their results (heatmap etc.) I propose to authors to adopt a more simplified language to share their interesting results, if we want the other researchers will be able to understand and use them as much as possible. For example, even the Venn diagram of figure 1 (which is usually an easy way for comparing and classifying results) is completely difficult to understand. Unfortunately, we do not find prompt explanations neither in figure legend, nor in the text. Therefore, I invite authors to provide clearer explanations in figure legends.  

I also propose that authors choose the most significant results and present related evidence as supplementary supports, to facilitate reading and understanding of the paper.  

I finish my comments by a technical question: why authors did not use an appropriate histological staining for fat tissues? Results of H&E staining are not usually as sharp as oil red techniques or other commonly used techniques which are more appropriate for fat vacuoles. Furthermore, normally for such kind of studies we use freeze section not paraffin embedding. Is possible to shed a light on choice of histological methods?

Author Response

Comments and Suggestions for Authors

In present study, researchers used an “extensive lipidomics analysis” to provide a better understanding of metabolic disorders and probable ways of curing or improving healing procedures by adding fish oil in routine diet.

This is one of the most complicated articles that I have ever read: there are many experimental groups which are compared to each other and to a large spectrum of lipids that have already been compared amongst each experimental group.

My first proposition to authors is to start simplifying the paper by the title.

Despite the complicated nature of “omics”, and they way of presenting their results (heatmap etc.) I propose to authors to adopt a more simplified language to share their interesting results, if we want the other researchers will be able to understand and use them as much as possible. For example, even the Venn diagram of figure 1 (which is usually an easy way for comparing and classifying results) is completely difficult to understand. Unfortunately, we do not find prompt explanations neither in figure legend, nor in the text. Therefore, I invite authors to provide clearer explanations in figure legends.

I also propose that authors choose the most significant results and present related evidence as supplementary supports, to facilitate reading and understanding of the paper.

I finish my comments by a technical question: why authors did not use an appropriate histological staining for fat tissues? Results of H&E staining are not usually as sharp as oil red techniques or other commonly used techniques which are more appropriate for fat vacuoles. Furthermore, normally for such kind of studies we use freeze section not paraffin embedding. Is possible to shed a light on choice of histological methods?

The title section has been streamlined, the plots and tables of the article have been annotated and explained, and the article has been modified in detail.

Results of H&E staining are not usually as sharp as oil red techniques or other commonly used techniques which are more appropriate for fat vacuoles.

Because due to H&E staining, various components of tissue cells can be colored, which is convenient for the observation of tissue structure after dietary intervention, and the staining of cells can be stored for a long time, mainly based on this consideration, and finally chose this method.

normally for such kind of studies we use freeze section not paraffin embedding. Is possible to shed a light on choice of histological methods?

Because paraffin embedding can be continuously sliced, the tissue structure is preserved intact, it is easier to observe the cell morphological structure, and the storage time is long, while the cryo-sectioning process is simpler than that of paraffin embedding, but the cell intact structure is more difficult to obtain and the tissue is fragile.

Reviewer 3 Report

The authors write that the number of metabolic syndromes is increasing. Metabolic syndrome is the designation of a status with dyslipidemia, hyperglycemia, central obesity, hypertension.  The plural is wrong and it is the number of patients with metabolic syndrome, not the number of metabolic syndromes which is increasing.

The English language is unsatisfactory and certain sentences do not make sense, like e.g. " due to the different symptoms of metabolic syndrome the funtion of different phospholipids is not the same ...". This sentence does not make any sense. The symptoms do not indluence the function of phospholipids

Author Response

Comments and Suggestions for Authors

The authors write that the number of metabolic syndromes is increasing. Metabolic syndrome is the designation of a status with dyslipidemia, hyperglycemia, central obesity, hypertension. The plural is wrong and it is the number of patients with metabolic syndrome, not the number of metabolic syndromes which is increasing.

The English language is unsatisfactory and certain sentences do not make sense, like e.g. " due to the different symptoms of metabolic syndrome the funtion of different phospholipids is not the same ...". This sentence does not make any sense. The symptoms do not indluence the function of phospholipids

In response to your question, I have rewritten and revised the article

Round 2

Reviewer 1 Report

Well revised. 
